# Oncogenic herpesvirus KSHV triggers hallmarks of alternative lengthening of telomeres

Timothy P. Lippert[1,2], Paulina Marzec [1], Aurora I. Idilli [1], Grzegorz Sarek[1], Aleksandra Vancevska[1], Mark Bower[3], Paul J. Farrell[4], Päivi M. Ojala [4,5], Niklas Feldhahn [2,6] & Simon J. Boulton [1,6✉]

To achieve replicative immortality, cancer cells must activate telomere maintenance mechanisms to prevent telomere shortening. ~85% of cancers circumvent telomeric attrition by re-expressing telomerase, while the remaining ~15% of cancers induce alternative lengthening of telomeres (ALT), which relies on break-induced replication (BIR) and telomere recombination. Although ALT tumours were first reported over 20 years ago, the mechanism of ALT induction remains unclear and no study to date has described a cell-based model that permits the induction of ALT. Here, we demonstrate that infection with Kaposi's sarcoma herpesvirus (KSHV) induces sustained acquisition of ALT-like features in previously non-ALT cell lines. KSHV-infected cells acquire hallmarks of ALT activity that are also observed in KSHV-associated tumour biopsies. Down-regulating BIR impairs KSHV latency, suggesting that KSHV co-opts ALT for viral functionality. This study uncovers KSHV infection as a means to study telomere maintenance by ALT and reveals features of ALT in KSHV-associated tumours.

[1] The Francis Crick Institute, 1 Midland Road, London NW11AT, UK. [2] Department of Immunology & Inflammation, Centre for Haematology, Du Cane Road, London W12 0NN, UK. [3] National Centre for HIV Malignancy, Department of Oncology, Chelsea & Westminster Hospital, Fulham Road, London SW10 9NH, UK. [4] Section of Virology, Department of Infectious Diseases, Imperial College London, Norfolk Place, London W2 1PG, UK. [5] Translational Cancer Medicine Research Program, University of Helsinki, Haartmaninkatu 8, Helsinki 00290, Finland. [6]These authors contributed equally: Niklas Feldhahn, Simon J. Boulton. ✉email: simon.boulton@crick.ac.uk

KSHV is the aetiological agent of Kaposi's sarcoma (KS), primary effusion lymphoma (PEL) and a proportion multicentric Castleman's disease (MCD)[1]. In such tumours, the virus persists in a latent state as multiple copies of a plasmid-like episome tethered to host chromatin by the key latency factor LANA[2,3]. Sub-cellular localisation studies of LANA have revealed that a subset of the protein is localised to subtelomeres or telomeres[4]. Furthermore, LANA has been reported to interact with the key telomere binding protein TRF1 upon ectopic overexpression[5], which could explain its telomeric localisation. During oncogenic processes, telomeric chromatin is often altered and is associated with the induction of the DNA damage response (DDR) at telomeres[6]. The host DDR machinery has also been shown to impact KSHV replication during latency[7–9] and multiple DDR, DNA replication and chromatin factors have been found associated with viral proteins in a systematic KSHV virus–host proteomic screen[10]. These observations led us to test the hypothesis that KSHV-encoded viral proteins engage with and modulate host telomeres and/or the DDR machinery during infection.

To this end, we investigate the impact of KSHV on telomeres in host cells in detail and find that infection triggers hallmarks of telomere elongation by homologous recombination (ALT) in initially non-ALT cell lines. Our data further suggest that intact BIR associated with ALT is important for the establishment of viral latency, but dispensable for KSHV genome synthesis. We further investigate whether this is the case in the most frequent KSHV-associated tumour, KS, and show that indeed, key hallmarks of ALT are present in such tumour tissue. These data identify a new tool to comprehensively intiate de novo hallmarks of ALT in culture and also show that KS exhibits features of a tumour with active ALT telomere maintenance.

## Results

**Latent KSHV infection alters the telomeric proteome in a manner consistent with telomere recombination and DNA damage.** To examine if telomere function is modulated in response to KSHV, we first interrogated how the proteomic composition of telomeres is altered in cells upon infection with rKSHV.219[11] (KSHV). Telomerase positive BJAB cell lines with stable latent viral infection were established by de novo KSHV infection as described previously[12] and were analysed at 30 days post infection in the absence of detectable lytic reactivation (Supplementary Fig. 1). To directly compare the telomeric proteome of uninfected cultured BJAB cells and KSHV+ BJAB cells, we performed Proteomics of Isolated Chromatin fragments (PICh)[13] on equal numbers of cells. Pulldowns with a 2 F'-RNA telomeric probe resulted in telomere-specific protein enrichment when compared to a scrambled control probe (Supplementary Fig. 2a). This was confirmed by mass spectrometry analysis, which revealed the enrichment of the six subunit telomere-bound protein complex named Shelterin[14] in both parental BJAB and KSHV+ BJAB cells with the telomere-specific probe, but not with the scrambled control (Fig. 1a, Supplementary Fig. 2b). A notable difference between parental BJAB and KSHV+ BJAB cells was the presence of DDR proteins at telomeres of KSHV+ BJAB cells including Mre11/Rad50/Nbs1, Fen1 and the RPA heterotrimer. We also observed SLX4, SLX4IP and orphan nuclear receptors (NR2C2, NR2C1, NR3C2) at telomeres of KSHV+ BJAB cells, which were largely absent from the telomeres of parental BJAB cells. The presence of DDR factors, SLX4, SLX4IP and orphan nuclear receptors at the telomeres of KSHV+ BJAB cells is reminiscent of telomeres undergoing recombination, including cells using the ALT pathway of telomere maintenance[15–18]. These data raised the possibility that KSHV infection may induce DNA damage and recombination at telomeres, potentially suggestive of ALT.

**KSHV infection triggers hallmarks of ALT activity in multiple cell lines.** ALT is a telomerase-independent mechanism of telomere maintenance employed in ~15% of all cancers to achieve proliferative immortality[19], which extends telomeres by recombination and break-induced replication (BIR)[20,21]. Cells undergoing ALT exhibit a number of characteristics including: telomere DNA damage, telomere–telomere recombination, clustering of recombining telomeres within PML bodies (APBs)[22], the presence of extra-telomeric DNA (such as C-circles)[23,24], which are by-products of the telomere recombination reaction, and telomere length heterogeneity. These hallmarks of ALT were absent from parental BJAB cells. However, KSHV+ BJAB cells presented with increased telomere clustering, elevated APB numbers and evidence of C-circles, which are the by-products of telomere recombination (Fig. 1b–e). These data strengthened the notion that KSHV infection may trigger an ALT-like mechanism in previously telomerase positive cells.

We next assessed if induction of ALT characteristics following KSHV infection is also observed in commonly used cell line models of KS, the main KSHV-associated cancer. To this end, we infected SLK and EA.hy926 cell lines with rKSHV.219[25]. To exclude any differences due to clonal selection, we established at least three independently derived KSHV+ SLK and KSHV+ EA.hy926 cell lines for subsequent analysis. We first examined whether KSHV infection leads to the induction of DNA damage in the form of telomere-induced damage foci (TIFs), which are evident at ALT telomeres[26]. Analysis of KSHV+ SLK and KSHV+ EA.hy926 cell lines revealed enrichment of DNA damage-induced phosphorylated RPA (pRPA) at telomeres, when compared to uninfected SLK and EA.hy926 parental cell lines (Fig. 1f, g). KSHV+ SLK and KSHV+ EA.hy926 cell lines also exhibited APBs (Fig. 1h, i), which are the sites of telomere clustering and recombination in ALT. Recombination between telomeres can be monitored by assessing the frequency of telomere sister-chromatid exchange (T-SCE) and the presence of telomeric recombination by-products in the form of C-circles[27,28]. In contrast to uninfected control SLK and EA.hy926 parental cell lines, KSHV+ SLK and KSHV+ EA.hy926 cells displayed significantly elevated levels of T-SCE (Fig. 2a, b) as measured by chromosome orientation FISH (CO-FISH). We also observed increased telomere fragility (Supplementary Fig. 3), consistent with heightened telomere damage and replication stress[29], which manifest as increased pRPA-positive TIFs specifically at telomeres in KSHV+ cells. Next, we measured the occurrence of C-circles using a Φ29-dependent rolling-circle polymerisation assay and telomere Southern blotting (Fig. 2c). Quantification of the signal obtained across biological replicates indicated that C-circles accumulate in KSHV+ SLK and KSHV+ EA.hy926 cells, but not in the uninfected parental controls (Fig. 2d). We also employed quantitative telomere FISH (qFISH) on interphase nuclei to further assess telomere clustering (Fig. 2e). KSHV-infected SLK and EA.hy926 cells exhibited telomere signal heterogeneity and a corresponding decrease in the number of individual telomere FISH foci per nucleus (Fig. 2f, g), which is indicative of telomere clustering. Analysis of telomere length by metaphase qFISH and terminal restriction fragment (TRF) Southern blotting confirmed that KSHV infection induces both telomere lengthening and size heterogeneity (Fig. 2h, Supplementary Fig. 4a). Direct comparison of ALT hallmarks in KSHV+ cells with established ALT cell line U2OS showed that these were induced to a lower, but comparable level (Supplementary Fig. 4b–d). Taken together, these data indicate that hallmarks of ALT are induced in cells subject to latent KSHV infection.

To exclude that the changes observed upon KSHV infection are due to telomerase hyper-activation rather than induction of ALT, we performed RT-qPCR to measure expression of the telomerase

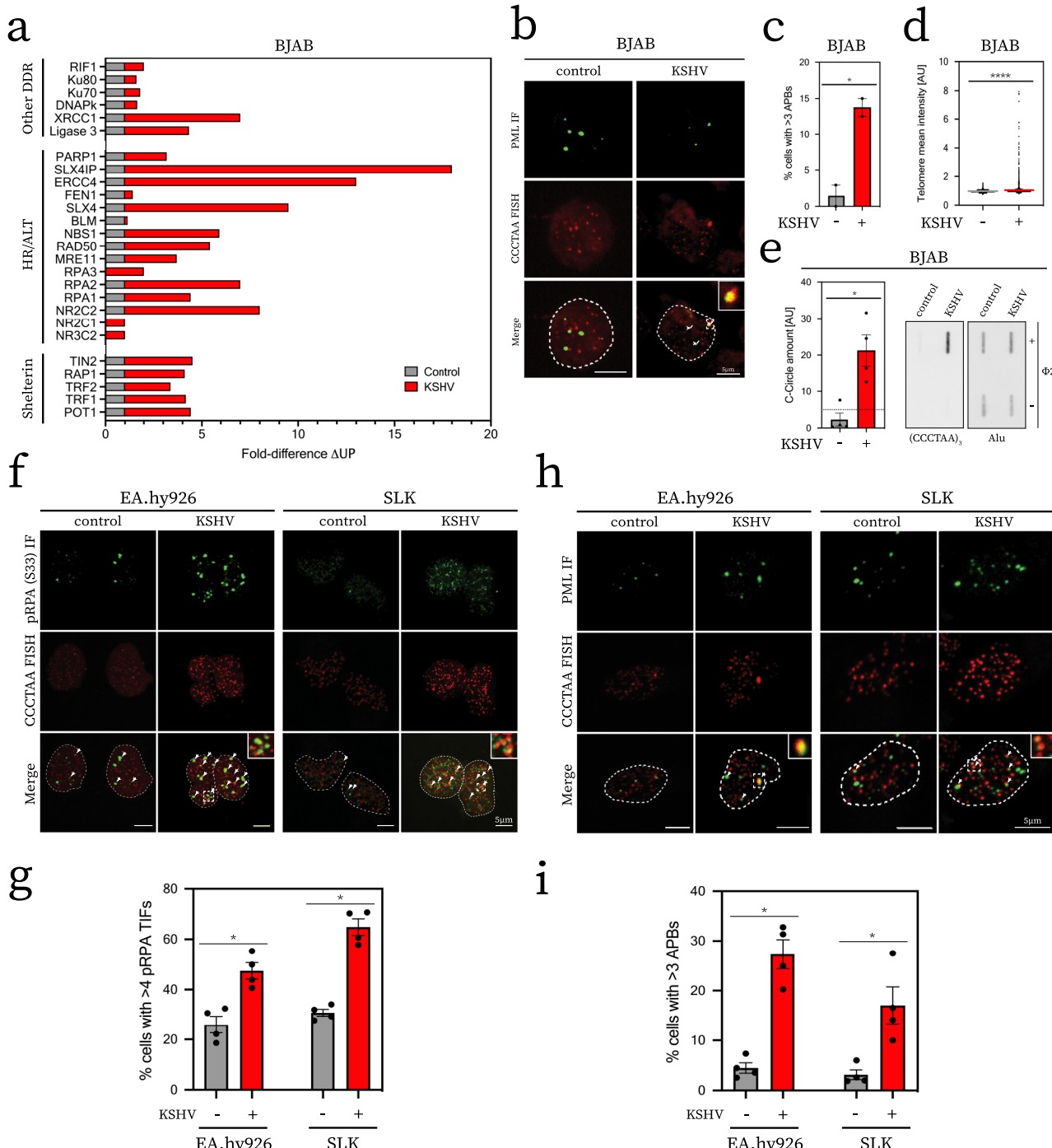

**Fig. 1 KSHV infection results in recruitment of DDR and recombination factors to telomeres. a** Analysis of telomere-associated proteome in response to infection. Data is presented as fold-difference in number of unique peptides relative to uninfected co-culture as determined by spectral peak calling. Peptides in scrambled pulldown were subtracted prior to comparison. **b** Occurrence of APBs in infected BJAB cells. Representative micrograph of the infected cell line and uninfected co-culture is shown. Arrows denote incidences of APBs in merged image, dotted line indicates outline of DAPI counterstain (not shown). **c** Quantification of cells with APBs. Experiment was performed in biological duplicate on the at least 50 cells from each cell line. Statistical significance was inferred by unpaired, two-tailed $t$-test (*$p = 0.0243$). **d** Quantification of mean telomere signal intensity. Telomere FISH signal intensity of cells included in analysis shown in **c** were quantified and plotted as dot plot. Significance was tested with an unpaired, two-tailed $t$-test (****$p < 0.0001$). **e** Quantification of signal obtained from C-circle analysis. Experiment was performed in technical quadruplicate and quantification was carried out by subtracting the $\Phi 29$-negative controls. Dotted line represents threshold, which determined positive samples (>5). Statistical significance was tested by unpaired, two-tailed Mann–Whitney test (*$p = 0.028$). Representative blot shown on the right next to quantification. **f** Representative confocal micrographs of pRPA TIF staining, inset presents digital zoom of area marked by dashed box in main image. Dotted line denotes the outline of DAPI nuclear counterstain (not shown) and arrows show pRPA TIFs which were counted in the analysis. **g** Quantification of occurrence of pRPA TIFs depicted in **f**. Experiment was performed in biological quadruplicate with 8 independently established KSHV positive cell lines. Each experiment analysed at least 50 cells per condition. Statistical significance was inferred by Mann–Whitney test (all *$p = 0.0286$). **h, i** Analysis of the occurrence of APBs in an analogous manner to pRPA TIF analysis **f**, **g** using the same biological quadruplicate, presentation style and statistical test. All error bars represent SEM.

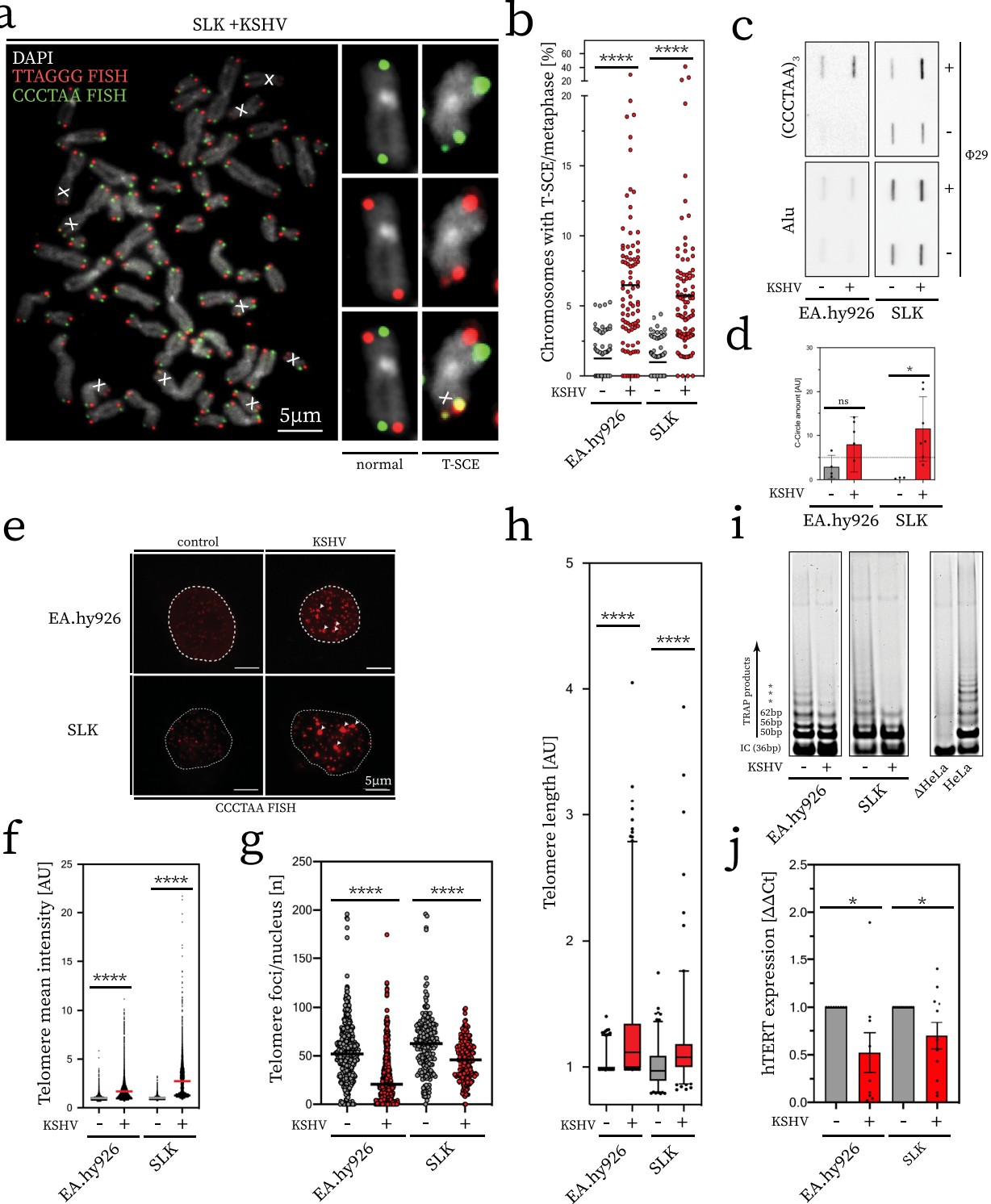

catalytic subunit *hTERT* and conducted telomere repeat amplification protocol (TRAP) analysis, which is a measure of telomerase activity. When compared to uninfected parental SLK and EA.hy926 cells, the corresponding KSHV-infected cell lines exhibited a decrease in *hTERT* expression as well as a reduction in telomerase activity by TRAP assay (Fig. 2i, j). These results are in contrast to previously published experiments using ectopic expression of KSHV LANA and telomerase promoter constructs[30,31]. Taken together, these data indicate that KSHV infection induces hallmarks of ALT and a concomitant down-regulation of telomerase.

**ALT BIR is important for the efficient establishment of KSHV latency but dispensable for viral genome replication**. In ALT tumour cells, telomere recombination and replication is achieved by an atypical break-induced replication (BIR) mechanism[20,21,32]. We first considered the possibility that cells infected by KSHV may exhibit a reliance on BIR for proliferation. To test this possibility, we examined the impact on cell growth of deleting BIR factors in uninfected parental cells versus KSHV-infected SLK cells with established latent infection. Depletion of BIR factors BLM, SLX4, PolD3, and RAD52 had no detrimental impact on the proliferation of uninfected SLK parental cell lines

**Fig. 2 KSHV⁺ cell lines exhibit hallmarks of telomere maintenance by ALT. a** Representative image of metaphases from KSHV-infected cells. Small panels show chromosomes with normal signal pattern (left column) and T-SCE (right), respectively. X symbol within image denotes T-SCE events on the metaphase depicted. **b** Blinded quantification of T-SCE/metaphase. Experiment was performed in biological triplicate (**a**). Statistical significance was tested with unpaired, two-tailed *t*-test (all ****$p < 0.0001$). **c** Representative C-circle telomere slot blot. Circular telomeric DNA was analysed upon amplification with Φ29 ( + ) as well as in its absence (−) as background control. Slot blot was probed for telomeric DNA (upper panel) and Alu probe as loading control (bottom panel). **d** Quantification of signal obtained from biological quadruplicate. Signal quantification was carried out by subtraction of Φ29-negative controls. Dotted line represents threshold, which determined positive samples in line with published reports (>5). Error bars represent SEM. Significance tested with two-tailed Mann–Whitney test (EA.hy926 ⁿˢ$p = 0.1714$, SLK *$p = 0.0006$). **e** Representative interphase telomere FISH. Telomere clustering is marked by white arrows in image. **f** Quantification of mean telomere signal intensity. Data from biological triplicate and statistical significance determined as in **b**. **g** Count of telomere FISH foci used for intensity measurement (**f**). Significance inferred in a manner equivalent to **b**, **f**. **h** Analysis of relative telomere length by metaphase qFISH. AU denotes relative telomere signal intensity, see associated Methods section. Experiment was performed in biological triplicate for each cell line, whiskers represent 1–99 percentile and box is median with two adjacent quartiles. Statistical significance was tested with Student's *t* test (all ****$p < 0.0001$). **i** Representative TRAP products. SYBR Gold stained gel image shown for each cell line, representative of biological triplicate, along with control conditions (HeLa, telomerase positive cell line, ΔHeLa heat inactivated lysate). **j** Expression of *hTERT*. RT-qPCR measurement is plotted as relative mean, normalised to uninfected control cell lines, respectively. Data obtained from three independent biological replicate cell lines, performed in technical triplicate. Statistical significance was tested with unpaired, two-tailed Student's *t* test (EA.hy926 *$p = 0.0351$, SLK *$p = 0.0442$). Error bars represent SEM.

in colony formation assays (Fig. 3a, b). In contrast, KSHV-infected SLK cells that exhibit all of the hallmarks of ALT as described above, show reduced proliferative capacity when subject to depletion of BIR factors (Fig. 3b, Supplementary Fig. 5a, b). Hence, the induction of ALT in KSHV-infected cells leads to a dependency on BIR for efficient cellular proliferation. Prompted by these results, we also explored the possibility that BIR might contribute to the establishment of KSHV latency. To this end, we depleted BIR factors by siRNA in parental SLK and EA.hy926 cells. Following knockdown of BIR factors these cells were then exposed to rKSHV.219[25] (Fig. 3c, d). Analysis of relative viral copy number by qPCR revealed that depletion of BIR factors resulted in a decrease in the levels of viral episomes when compared to control siRNA (Fig. 3e). These data suggest that loss of BIR in the host cell reduces the ability of KSHV to establish latency upon de novo infection.

The mechanism of KSHV latent DNA replication is complex and may occur outside of bulk S-phase DNA synthesis in infected cells[33]. Given that depleting BIR reduces the efficiency of latency establishment and results in lower levels of viral episomes, we examined whether BIR might impact on KSHV latent DNA replication. To this end, we modified an existing BrdU pulldown method used to purify DNA synthesised by BIR[20] and quantified the occurrence of KSHV genomes by ddPCR. Applying this method to cells arrested in G2/M allowed us to retrieve viral DNA in BrdU pull-downs relative to IgG control (Supplementary data Fig. 7a, b). These data indicate that KSHV genomes can be synthesised during G2/M where a BIR-like mechanism may contribute to viral DNA synthesis. To determine whether KSHV replication is limited to G2/M, we compared KSHV sequence enrichment in BrdU pull-downs during G2/M and S phase. BrdU is efficiently incorporated into KSHV genomes during both cell cycle phases (Supplementary Fig. 7c, d). Next, we tested whether knockdown of BIR factors decreases KSHV DNA synthesis in G2. Our results showed that loss of BIR factors does not impede viral DNA replication in G2 (Fig. 3f). These data show that although BIR is required for survival of KSHV-infected cells and necessary for the establishment of viral latency, it is dispensable for KSHV replication and copy number maintenance.

Next, using LANA as a proxy for latent KSHV episomes tethered to host chromatin, we examined the coincidence of LANA foci and BIR sites visualised by pulse-labelling of cells arrested in G2/M with EdU (Supplementary Fig. 8a, b). Click-iT chemistry and fluorescent microscopy[34] revealed a subset of

LANA foci colocalised with sites of active DNA synthesis in KSHV-infected cells (Fig. 3g). Notably, a subset of LANA/EdU foci overlapped with telomeres (Fig. 3g, h), suggesting that KSHV episome replication occurs at sites of active BIR in cells. Knockdown of BIR factors decreased the occurrence of LANA/EdU foci but did not decrease the number of LANA foci or KSHV genomes overall at the time point of analysis (Fig. 3h, Supplementary Fig. 8c, d). Furthermore, LANA-associated BIR foci were found to colocalise with telomeres in 10% (SLK cells) and 14% (EA.hy926 cells) of cases, consistent with subset of LANA existing at sites of ALT BIR (Fig. 3i). Collectively, these observations suggest that BIR contributes to viral latency, potentially facilitating efficient viral DNA synthesis. These data also provide a potential explanation for previously observed HR intermediates at KSHV episomes during latency[33].

**Features of ALT upon infection are not due to loss of ATRX, DAXX or ASF1.** The histone H3.3 chaperones ATRX and DAXX are commonly deleted in ALT tumours[35] and replication stress at telomeres resulting from their loss may contribute to the induction and/or maintenance of ALT[36]. Importantly, however, loss of ATRX and/or DAXX is insufficient to trigger ALT de novo and not all ALT⁺ cells exhibit loss of these proteins[35,37]. In contrast, transient depletion of the essential histone chaperone ASF1 is the only known trigger of ALT in cell culture[38]. We therefore examined whether downregulation of these factors underlies ALT induction upon KSHV infection. While ATRX protein level differs between parental SLK and EA.hy926 cell lines, the levels of ATRX and ASF1 remained similar in KSHV-infected cells when compared to uninfected controls (Supplementary Fig. 9a). Further analysis of these factors by RT-qPCR revealed that there was a small but significant decrease in the expression of ATRX and ASF1a (Supplementary Fig. 9b). DAXX protein and mRNA levels were slightly elevated in SLK cells only, but importantly no loss of DAXX was detected in either cell line upon infection (Supplementary Fig. 9a, b). In summary, these experiments indicate that there is no complete loss of ATRX, ASF1a or DAXX, indicating that this mechanism is unlikely to cause ALT induction upon KSHV infection. However, it remains possible that KSHV modulates the function of these factors during ALT induction, independent of their expression.

**Kaposi's Sarcoma tumours show features of ALT activity.** To investigate the clinical relevance of our findings, we examined whether features of ALT are evident in tumour biopsies from 40

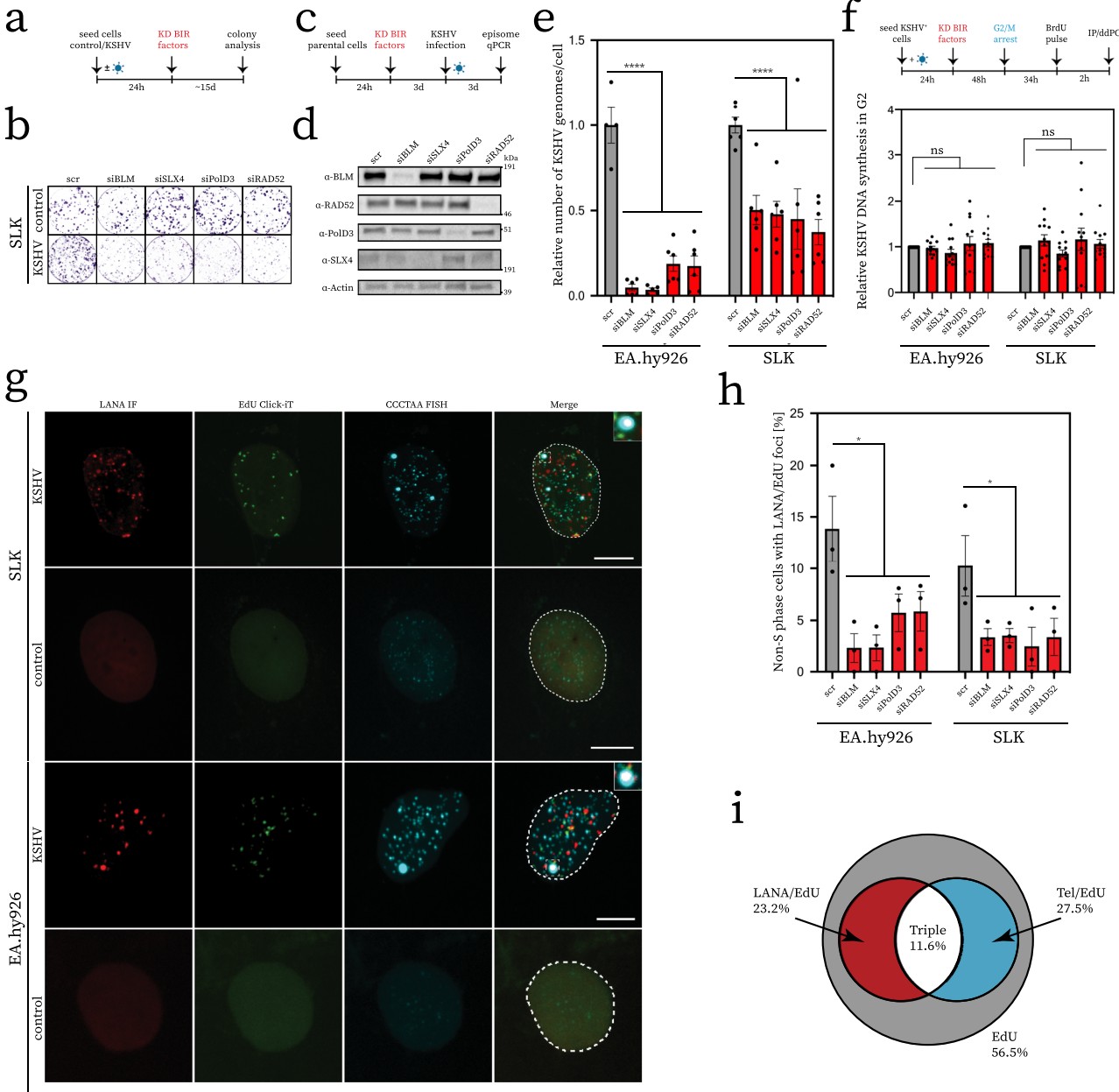

**Fig. 3 ALT BIR facilitates efficient viral latency. a** Schematic overview of Clonogenic assay. **b** Representative images of Clonogenic assay. See Supplementary Fig. 5 for quantification across biological replicates. **c** Schematic overview of experiment performed in **d**. **d** Depletion of key ALT factors by RNA interference. Cells were treated with siRNA indicated and protein levels were determined at time point of analysis by Western blot with antibodies as indicated. **e** Quantification of the relative number of KSHV genomes/cell upon depletion of key ALT factors. qPCR was performed against a standard curve of fixed amplicon molecules across conditions to infer copy number. Data presented relative to number determined for scrambled siRNA treatment. Experiment performed in biological duplicate, technical triplicate; Statistical significance tested by ordinary one-way ANOVA using scrambled siRNA condition as control (EA.hy926 ****$p < 0.0001$, SLK ****$p = 0.0014$). **f** Schematic overview of BrdU pulldown experiment (top). Results of experimental quadruplicate measurement by ddPCR upon knockdown of candidate ALT factors. Data expressed as relative KSHV amplicon number normalised to housekeeping control reaction. Statistical significance tested by one-way ANOVA (EA.hy926 $^{ns}p = 0.4246$, SLK $^{ns}p = 0.4777$). **g** Representative micrograph showing virus-associated BIR foci. Images show a maximum intensity Z-projection of confocal images. **h** Quantification of the occurrence of virus-associated BIR foci (LANA/EdU co-localisation) upon knockdown of key ALT factors. Cells arrested in G2/M were considered. Any residual S-phase cells were discounted from analysis as evident by strong pan-nuclear EdU stain. Data representative of three biological replicates, at least 100 cells were included in the analysis of each condition. Statistical significance was inferred by one-way ANOVA (*$p < 0.05$, ns, $p = 0.069$). Error bars in **e**, **f**, **h** represent SEM. **i** Venn diagram showing overlap of signals considered in analysis depicted in **g**, **h**. Percentages were derived from three independent experiments, scrambled control condition only.

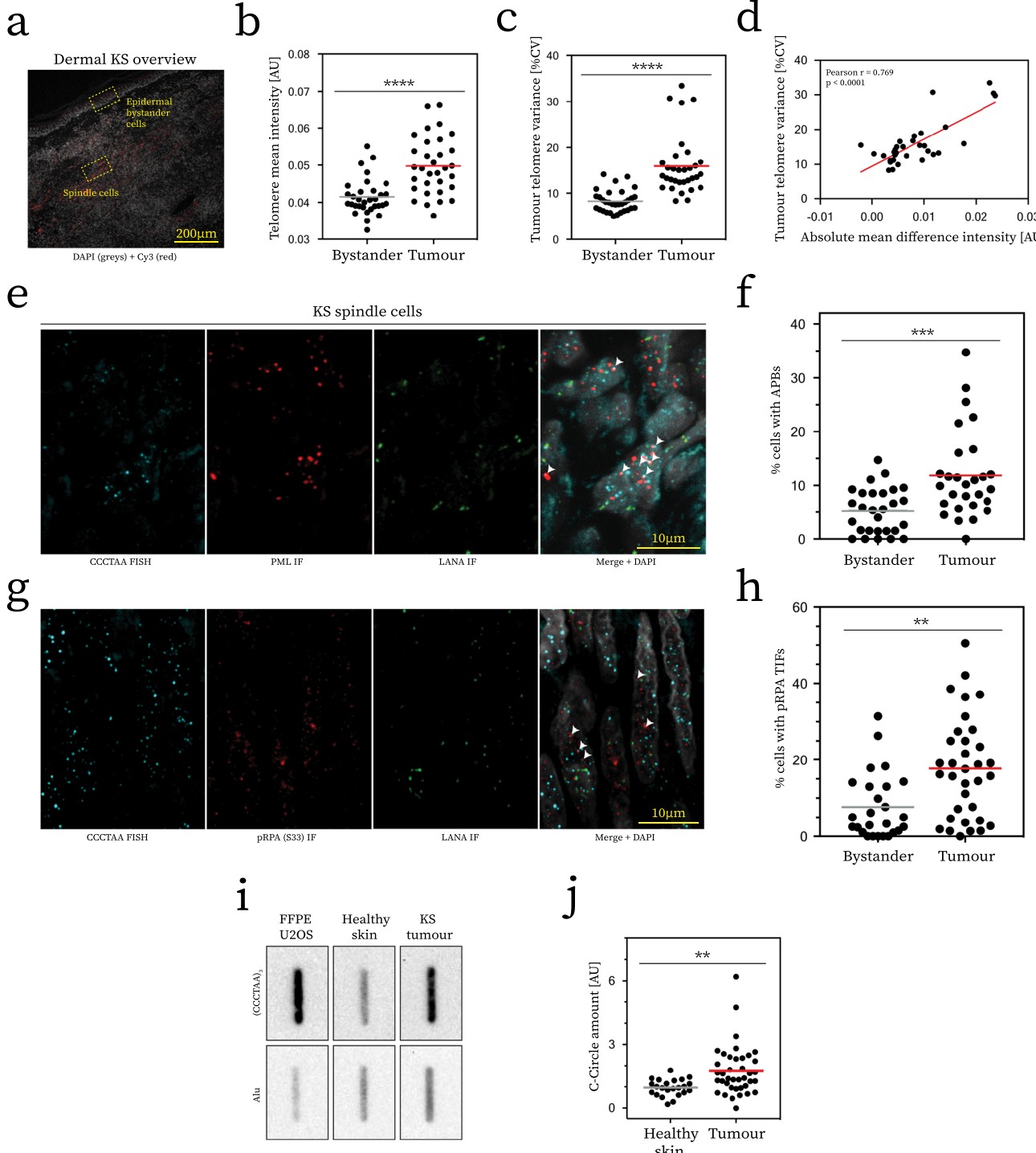

cases of dermal KS, the main KSHV-associated cancer. To normalise telomere analysis within each patient biopsy, we compared KS spindle tumour cells with adjacent normal epidermal bystander cells present within the margins of the biopsy (Fig. 4a), which are readily identifiable as KSHV negative due to the absence of LANA staining (Supplementary Fig. 10a, b). qFISH analysis revealed a consistent increased mean telomere signal intensity and heterogeneity in KS tumour cells relative to adjacent epidermal bystanders (Fig. 4b, c, Supplementary Fig. 12a), which surpassed cell type-intrinsic differences in 60% of the cases (Supplementary Fig. 11a, b). Moreover, we observed a strong correlation between mean increase in signal intensity and

telomere length heterogeneity between KS tumour cells and adjacent epidermal bystanders (Fig. 4d), consistent with telomere maintenance by ALT specifically in KS cells.

Next, we evaluated KS tumour material for telomeric pRPA and APBs by IHC/FISH. Again, we compared KS cells with adjacent epidermal bystander cells (Supplementary Fig. 11c, d). Consistent with the presence of ALT, we observed increased levels of APBs and pRPA at telomeres in KS tumour cells (Fig. 4e–h) but not in adjacent bystander cells. Finally, we isolated DNA from FFPE-embedded tumour tissue and subjected it to analysis for the presence of C-circles. We note that C-circles may not be adequately preserved in archived FFPE tissue, which may result

**Fig. 4 Dermal Kaposi's sarcoma tumours show features of ALT activity. a** Representative overview of dermal KS biopsy section. Micrograph shows DAPI nuclear counterstain in grey, autofluorescence of erythrocytes and extracellular matrix of tissue is visible in red (Cy3) for illustrative purposes. **b** Quantification of mean telomere signal intensity. Telomere FISH signals from every tumour cell nucleus and bystander epidermal cell within the same section were quantified, majority of cases >1000, derived from at least 100 cells per condition. Each dot represents a distinct patient, $n > 20$. Statistical significance was tested with paired Wilcoxon signed-rank test (****$p < 0.0001$). **c** Quantification of variance of data obtained by mean signal intensity measurements. Coefficient of variance was determined for each cell type and plotted, analysis outcome and smaple size as in **b**. **d** Plot showing correlation of measurements presented in **b**, **c**. Correlation was deteremined with Pearson's correlation coefficient (PCC). Line of best fit generated by linear regression (slope = PCC value). **e** Representative IHC/FISH image of spindle cells showing LANA and PML IHC staining and telomere FISH signals used for analysis in **b**–**d** generated by maximum intensity projection in Z of confocal micrographs. **f** Comparative quantification of proportion of cells with APBs (1/3 PML bodies at telomeres) for tumour cells and epidermal bystander cells within the same tissue section. Significance tested by paired Wilcoxon signed-rank test ($n > 20$, ***$p = 0.0008$). **g** Representative image generated in same manner as described for **e**, except for IHC was performed for pRPA (S33) instead of PML. **h** Quantification of cells with pRPA TIFs (at least one co-localisation event) for same cell types as in **f** ($n > 20$, **$p = 0.0013$). **i** Representative southern blot image of C-circle analysis. Slot blots were hybridised with probes specific to telomeres (top row) and Alu repeat for loading control (bottom row). **j** Quantification of C-circles in primary tumour material normalised to loading control. Each dot represents one patient or healthy skin donor, $n > 20$ each. Significance tested by unpaired, two-tailed $t$-test (**$p = 0.0026$).

in a false negative result for some patients. Nevertheless, we detected an increase in C-circles in 33% of tumour tissue samples, which was significantly elevated in comparison to samples from a cohort of healthy skin donors (Fig. 4i, j; Supplementary Fig. 12b). We observed a positive correlation between measurements employed here to infer ALT activity in the tumour material (Supplementary Fig. 12c), which is in line with ALT activation in KS primary tumour material.

## Discussion

In summary, we present evidence that KSHV infection in human cells triggers the induction of telomere maintenance by ALT. We show that intact ALT-associated BIR contributes to KSHV latency, which raise the possibility that KSHV induces ALT in order to co-opt the BIR machine for efficient viral functionality. KSHV DNA synthesis is not limited to G2/M and is still largely intact upon knockdown of BIR factors. Because the KSHV episome is difficult to replicate due to abundant GC-rich sequences and prone to DNA secondary structures such as G4 quadruplexes[39], we speculate that ALT BIR may serve as an important repair pathway in response to KSHV episome replication fork collapse. Another possible explanation is that KSHV episomes tethered to telomeres are more susceptible to DNA breaks and thus rely on BIR for their effective repair. Such a dependency would also explain decreased survival of KSHV-infected cells in the absence of BIR since extensive, unrepaired DNA damage could lead to impaired proliferation. Thus, our data raise the hypothesis that ALT induction reported here is a by-product of such viral functionality in proximity to telomeres. These findings may also be applicable to infection with related Epstein-Barr Virus (EBV), since it is able to transiently induce a subset of ALT-like features in host cells[40,41]. Prior to this study, the only report of comprehensive induction of ALT in cell culture occurred upon knockdown of the histone chaperone ASF1[38]. However, since ASF1 is essential and cells rapidly die, modulating ASF1 cannot be used to induce a durable change to ALT. Hence, the ability to employ KSHV infection to induce hallmarks of ALT in cell culture has the potential to transform our understanding of the establishment, mechanisms and vulnerabilities underpinning ALT. To this end, it is crucial to determine the relevance of residual telomerase activity in the onset and/or maintenance of ALT since both KSHV-induced ALT cells and ALT induction upon ASF1 knockdown exhibit this property. In addition to this, determination of which virally encoded KSHV latency factor(s) are necessary for ALT induction and how these interact and modulate the host cell may illuminate how ALT is induced and could be targeted in sporadic cancers. Finally, we suggest that ALT hallmarks upon KSHV infection are not limited to cell

culture but it also evident in human patients with KS, which we implicate here as a novel subtype of ALT cancer.

## Methods

**Human tissue**. The normal human skin paraffin embedded samples used as part of this research were fully anonymised and collected before the Human Tissue Act 2004 was implemented in September 2006. These are stored under the Francis Crick Institute's HTA license (#12650) (originally donated by NHS Whittington Hospital to the Experimental Histopathology Science Technology platform at the Francis Crick Institute). They are classed as existing holdings, therefore were able to be used for research without the need for consent. Archived FFPE-embedded tissue sections from KS patients are continuously collected, anonymised and deposited in the Imperial College Healthcare Tissue Bank (ICHTB) under supervision of M.B. ICHTB is approved by the National Research Ethics Service (NRES) to release human material for research. The samples for this project are issued from a sub-collection that covers this project and has been approved by ICHTB (HTA #12275).

**Cell lines**. All cells were cultured under culture conditions recommended by ATCC. SLK and iSLK.219 producer cell lines were obtained from Dr Grzegorz Sarek, and initially generated by Dr Don Ganem, as published[25]. BJAB cell lines were obtained from Prof. Thomas Schulz. Uninfected EA.hy926 cells were purchased from ATCC. STR profiling was performed to confirm identity of all cell lines and mycoplasma testing was carried out by Cell Services STP (Crick Institute).

**Generation of KSHV positive cell lines**. iSLK rKSHV.219 producer cells[25] were used to generate infectious KSHV virions. At 60% confluency, lytic replication was induced with 1 µg/mL Doxycyclin and 1.2 mM Sodium butyrate for 72 h. The supernatant (7 ml in 10 cm dish) was cleared by centrifugation at 500 $g$ for 5 min and drawn through a 0.45 mm filter. SLK or EA.hy926 cells were seeded into 6-well plates at 60,000 cells per well on the day prior to infection. 2 mL of the resulting infectious medium was applied to target cells in complete medium supplemented with 10 µg/mL Polybrene. Infection was enhanced by spinoculation at 2300 rpm, 32 °C for 1 h 30 mins (Acceleration 3, Deceleration 1) and then incubated overnight at 37 °C. Infectious medium was replaced with fresh medium the next day. Stable 100% KSHV positive cell lines were obtained by selection with 4 µg/mL puromycin. All analysis of KSHV-infected cell lines generated in this study was carried out at >30 days post infection.

**Proteomics of isolated chromatin fragments (PICh)**. BJAB cells were crosslinked in 3.6% formaldehyde/PBS for 30 min and processed using PICh[15] with modifications: $10^9$ cells per pulldown were used. Cells were sonicated in 50 mM Tris pH8, 10 mM EDTA, 200 mM NaCl, 1% SDS, desalted and hybridised with 5 µL of 100 µM 2'-F RNA telomere probe (Desthiobiotin-108 Carbons-5'-(UUAGGG)7.5) or scrambled probe (Desthiobiotin-108 Carbons- 5'- (GAUGUG)7.5) and pulled down with 500 µL of streptavidin beads.

**Combined immunoflourescence–fluorescence in situ hybridisation (IF-FISH)**. Cells were grown on glass cover slips. Cytoplasm was pre-extracted by placing cover slips on ice in pre-chilled 0.5% Triton X-100/PBS for 5 min. Immunofluorescent staining was carried out as published[15] using the following antibodies: α-PML (Santa Cruz sc966 1:500), α-RPA (P-S33, Bethyl A300-246A 1:2,000), α-LANA (Millipore MABE1109, 1:250), α-mouse-Alexa647 (Invitrogen A21236 1:500), α-rat-Alexa488 (Invitrogen A1106 1:500), and α-rabbit-647 (Invitrogen A21245, 1:500). After staining, three washes with PBST were carried out followed by re-fixation in 2% Methanol-free Formaldehyde for 5 min. Telomere probe

(TelC-Cy5, PANAGENE #F2001) was applied at a final concentration of 4 nM as described[42].

**Chromosome-orientation FISH (CO-FISH)**. Incorporation of nucleoside analogues was achieved by addition of 7.5 μM BrdU/2.5 μM BrdC to complete growth medium for 18 h (SLK and EA.hy926 cell lines). Cells were arrested in metaphase with 0.1 μg/mL colcemid for 2 h. Metaphase spreads were produced as described[42]. Slides were treated with 0.5 μg/mL RNASe A in PBS for 30 min at 37 °C and stained with 10 μg/mL Hoechst 33258 in 2X SSC for 15 min. Single-stranded nicks were introduced by exposure of slides in a shallow bath of 2X SSC to UV light (365 nm) for 1 h. 10U/μL of Exonuclease III (Promega #M1811) was applied in buffer supplied with the kit at 37 °C twice for 1 h each. The slides were dipped briefly in PBS, H₂O and then left to dry completely overnight. The first probe (TelG-Alexa488, PANAGENE #F1008) was applied at 4 nM in hybridisation buffer (10% Dextran Sulfate, 50% deionized Formamide, 2X SSC) for 1 h 30 min. Slides were briefly washed in modified Wash #1 (10 mM Tris-HCl pH 7.5, 1 mg/mL BSA, 70% Formamide) and second probe was applied (TelC-Cy5, PANAGENE #F1003) at 4 nM in the same hybridisation buffer for 1 h 30 min. Slides were washed twice in Wash #1 (10 mM Tris-HCl pH 7.5, 1 mg/mL BSA, 70% Formamide) for 10 min followed by two washes in 2X SSC at 37 °C for 10 min. Finally, slides were three times in Wash #2 (100 mM Tris-HCl pH 7.5, 150 mM NaCl, 0.08% Tween-20) for 5 min, dipped in H₂O, air dried, and a cover slip was mounted using ProLong Gold with DAPI.

**Combined immunohistochemistry-fluorescence in situ hybridisation (IHC-FISH)**. Slides were heated at 65 °C for 10 min and washed twice for 3 min in Xylenes (mixture of isomers). Subsequently, slides were washed in the following sequence (3 min each): Xylenes:Ethanol 1:1, twice in 100% ethanol, 85% ethanol, 70% ethanol, 50% ethanol, cold water. Antigen retrieval was carried out in 10 mM Sodium Citrate, 0.05% Tween-20 pH 6. Submerged slides were heated in microwave at 700 W for 10 min and allowed to cool for 10 min. This was repeated once. Subsequently, slides were immersed in water and then washed twice for 3 min each in 70%, 85%, 100% ethanol. Slides were dried completely and hybridisation mix was applied as described for IF-FISH. Denaturation was carried out at 84 °C for 3 min. Hybridisation was >2 h at room temperature or overnight at 4 °C. This was followed by two washes in Wash #1 (10 mM Tris-HCl pH 7.5, 1 mg/mL BSA, 70% Formamide) for 15 min and three washes in PBST for 5 min each. Subsequently, antibodies were applied as described for IF-FISH.

**BIR foci analysis by EdU Click-iT chemistry**. BIR foci analysis was carried out as described with modifications[34]. Cells were enriched in G2/M phase of cell cycle by sequential treatment with 2 mM Thymidine (Sigma #T9250) for 16 h, recovery in normal medium for 6 h and treatment with Cdk1 inhibitor RO-3306 (Sellekchem #S7747) for 12 h. During the final 1 h of RO-3306 treatment, medium was supplemented with 10 μM EdU. For flow cytometry, cells were fixed in 4% Formaldehyde/PBS for 15 min at room temperature. The cells were washed three times with PBS followed by permeabilization via gentle addition of pre-chilled 100% methanol and incubation overnight at −20 °C. The cells were washed once with PBS. Alexa488 was coupled to incorporated EdU by Click-iT chemistry using Click-iT Plus imaging kit (ThermoFisher #C10637) following the manufacturer's protocol. Finally, cells were counterstained with DAPI at 0.2 μg/mL for 10 min. For cells grown on cover slips, the procedure outlined above was followed by incubation/washing cover slips in a 12-well plate followed by the IF-FISH procedure described earlier.

**BrdU IP analysis**. Cells were synchronised in G2 as described for BIR foci analysis or and incubated for 2 h with 100 μM BrdU. For the S phase analysis cells were synchronised by double thymidine block (2 mM thymidine for 16 h followed by 8 h release and 16h in 2 mM thymidine) and incubated for 2 h with 100 μM BrdU. DNA was isolated by Phenol-chloroform extraction following ON incubation with proteinase K at 55 °C. 10 μg of DNA was resuspended in 200 μL of 10 mM HEPES-KOH pH 7.9, 100 mM NaCl, 1 mM EDTA, 0.5 mM EGTA, 0.1% sodium deoxycholate, 0.5% sodium lauroylsarcosine and sonicated with Bioruptor (3 cycles 30 sec ON/30 sec OFF). DNA was denatured at 95 °C for 10 min and immediately cooled on ice. 20 μL of 10% triton was added to the sample and 0.5 μg of either mouse IgG or anti-BrdU antibody (BD Biosciences #347580). Samples were incubated overnight at 4 °C with the antibody and 3 h to overnight with 30 μL of protein G beads blocked with 0.5% BSA/PBS. Beads were washed 4 times in 50 mM HEPES-KOH pH 7.55, 250 mM LiCl, 1 mM EDTA, 1% NP-40, 0.7% sodium deoxycholate and once in TE/50 mM NaCl. DNA was eluted from the beads for 1 h shaking at 65 °C in 200 μL of 50 mM Tris-HCl pH 8, 10 mM EDTA, 1% SDS. DNA was precipitated with ethanol-sodium acetate and the pellet was washed 3 times in 70% EtOH, dried and resuspended in 50 μL of water. 2 μL of the sample was used for qPCR or ddPCR analysis.

For BrdU- IP immunoblot, 25uL of the sample and 5% of the input DNA was denatured for 10 min at 99 °C in 0.4 M NaOH, 10 mM EDTA and applied to nitrocellulose membrane through a slot-blot manifold. The membrane was rinsed in 2xSSC, UV crosslinked, blocked in 5% milk for 30 min and incubated for 2 h at RT with 2 mL of anti-BrdU antibody (GE #RPN202) and 40 min with anti-mouse HRP antibody.

**Microscopy image acquisition**. Metaphase images for CO-FISH analysis were acquired using a Nikon Eclipse E400 wide-field microscope equipped with a Hamamatsu Orca camera using a 100× oil-immersion objective. Images of live cells in culture for analysis of lytic cycle were obtained with an EVOS FL using a 10x objective. All other images were obtained using an inverted Nikon TiE microscope with a spinning disk module (Yokogawa CSU-X1).

**Microscopy image analysis**. Automated quantification of the mean intensity for each nuclear telomere FISH signal was carried out. To this end, multichannel greyscale images (telomere FISH, DAPI counterstain) were automatically imported and split into single channels using a custom FIJI[43] macro. Resulting image pairs were imported into CellProfiler[44] and analysed. Specifically, the pipeline created a binary mask for nuclei by thresholding the DAPI counterstain greyscale image. Signal corresponding to telomere FISH was filtered to enhance edges of foci and a second binary mask was generated by thresholding using Otsu's method[45]. Finally, telomere FISH signals, as determined by thresholding, were filtered on the basis of the nuclear outlined generated in previous step to exclude any extranuclear signals. The resulting mask, which only included nuclear telomere FISH signals, was laid over the raw image of the signal and quantification of signal intensity was achieved by determination of mean intensity for each focus.

Analysis of co-localisation in cell lines and tumour sections was performed as follows: Greyscale images corresponding to maximum intensity Z-projections were generated for each channel of interest and imported into CellProfiler[44] for automated analysis of co-localisation events. To achieve this, nuclei and telomeres were detected as described in the previous paragraph. Foci corresponding to IF staining were enhanced in analogy to telomere thresholding and a binary mask was generated. Masks corresponding to telomere FISH were overlaid with IF foci of interest to determine co-localisation, >30% overlap was deemed significant. Finally, in the case of images obtained from analysis of primary tumour material by IHC-FISH, an additional step to identify LANA-positive tumour cells was included by including only cells with LANA foci. In some experiments, automated image analysis was not possible due high variation in signal intensity, which impeded on accurate thresholding by one pipeline across all conditions (IF-FISH in cell lines) or complexity of the analysis task (BIR foci, CO-FISH). In these cases, image identity was scrambled computationally and images were scored manually.

**Telomere repeat amplification protocol (TRAP)**. 2000 cells were lysed on ice in 1X CHAPS buffer (10 mM Tris-HCl pH 7.5, 1 mM MgCl₂, 1 mM EGTA, 0.1 mM Benzamidine, 5 mM β-mercaptoethanol, 0.5% CHAPS, 10% Glycerol) for 30 min. Lysate was cleared by centrifugation at 15,000 x g for 15 min. TRAP reactions were set up as described in the manufacturer's protocol (Millipore #S7700) with the following modifications: i) Lysate was incubated with substrate for 1 h at 25 °C prior to PCR; ii) Samples were electrophoretically fractionated in 1X TBE on a pre-cast non-denaturing TBE 10% polyacrylamide gel; iii) Gel was stained with 1X SYBR Gold (Invitrogen #S11494)/1X TAE for 30 min followed by triple de-stain in 1X TAE, 5 min each.

**siRNA knockdown**. RNAi was performed as published previously[15].

**Clonogenic assay**. 500 cells per well were seeded into a 6-well plate and grown for 10–14 days until defined colonies were visible in untreated control conditions. Cells were fixed using 4% formaldehyde in PBS for 10 min. Formaldehyde was washed out with fresh PBS and colonies stained in 0.5% Crystal Violet for 1–2 h. Excess stain was removed under running water and plates dried completely for at least 24 h. Colony counts were obtained by automated imaging and analysis using a GelCount system (Oxford Optronix). For analysis by photospectrometry, stain was dissociated from wells by addition of 1% SDS/PBS and incubation on an orbital shaker for 1 h. The resulting solution was transferred to optical cuvettes and absorbance was measured at 600 nm.

**Episome copy number analysis by standard curve qPCR**. Uninfected SLK/EA. hy926 cells were subjected to siRNA knockdown as outlined above. On day 2 post transfection, cells were plated into 12-well plates at 50,000 cells per well. On day 3, cells were infected with rKSHV.219 as described in paragraph on the generation of KSHV-infected cell lines above. The virus was removed on the following day and cells were incubated for three days afterwards to allow for effective latency establishment. On day 6 post transfection, e.g. day 3 post infection, plates were washed with PBS and cell harvested. A standard curve of fixed number of amplicons obtained by standard PCR using primers specific to KSHV OriA region (All primer sequences in Supplementary Table 1)[46] was calculated using Avogadro's constant. The same primers were used for qPCR of the standard curve together with 100 ng of isolated DNA per condition. Signal was generated by amplification with SYBR Green JumpStart Taq ReadyMix (Sigma #S4438).

**Episome copy number analysis by ddPCR**. DNA samples were prepared as described in BrdU IP analysis. 2 μL of 10% input or BrdU IP was used per reaction. Amplification of KSHV OriA 205 bp region was multiplexed with amplification of a 200 bp region in housekeeping gene RPP30 in order to establish relative KSHV copy number per cell. All primers and TaqMan probes are listed in Supplementary Table 1. ddPCR was run in ddPCR Supermix for Probes (no UTP) (BioRad #1863023) with 0.9 μM of each primer and 0.25 μM of each probe (95 °C for 10 min, 40x (94 °C for 30 s, 62 °C for 60 s, 72 °C for 30 s); 72 °C for 5 min, 98 °C for 10 min).The absolute number of KSHV and RPP30 molecules in each 20 μL ddPCR reaction was determined using a ddPCR droplet reader and analysed with QuantaSoft software (BioRad). To determine KSHV copy number changes upon BIR knockdown, the absolute number of KSHV amplicons in each 20 μL reaction was normalised to the absolute copy number of RPP30 in the same 20 μL reaction. To analyse changes in KSHV replication upon BIR knockdown by BrdU IP, KSHV to RPP30 ratio from the BrdU IP was normalised to KSHV to RPP30 ratio in the input and compared to the scr siRNA control.

**RT-qPCR analysis of hTERT expression**. For each reaction, the equivalent cDNA of 100 ng RNA was used and expression of the gene of interest was achieved by usage of specific primers and the deltadelta $C_t$ method as published[47–49]. Primer sequences found in Supplementary Table 1.

**TRF Southern blot**. 10 μg of gDNA was incubated overnight at 37 °C with RsaI/HinfI, 15U each per reaction. The enzymes were then heat inactivated at 80 °C for 20 min and the sample was separated on a 1.5% w/v agarose/TAE gel. DNA in the gel was transferred on to positively charged Nylon membrane by capillary transfer and crosslinked by exposure to UV light (120 mJ/cm$^2$ UV) prior to hybridisation.

**C-Circle assay (CCA)**. C-Circle assay (CCA) was essentially performed as described[50]. Genomic DNA extraction from FFPE tumour tissue was performed using the QIAamp DNA FFPE Tissue Kit (Qiagen #56404). 2–4 five-micrometre-thick sections were scraped from microscope slides, deparaffinized in xylene and rehydrated with 70% ethanol. DNA concentration was measured by Qubit dsDNA HS Assay (Thermo Fisher Scientific # Q32854), and 15–50 ng of tumour genomic DNA was used to perform CCA rolling circle amplification reaction.

The CCA products were slot-blotted onto 2x SSC-soaked positive nylon membrane. After UV-crosslinking, the membrane was hybridised with telomeric probe, stripped and hybridised with Alu probe (details on probes described in subsequent section). The membrane was stripped by incubation in 50 mM NaOH for 30 min at 45 °C and followed by 50 mM Tris-HCl pH 7.5, 0.1X SSC, 0.1% SDS for 15 min. The result of the CCA was calculated as the signal intensity of the reaction with polymerase (+Φ29) divided by the amount of Alu repeats signal detected. A tumour sample was considered as ALT-positive, if the CC abundance was at least 2-fold above the background as published[50].

To evaluate CC status in KSHV-infected cell lines, CCA rolling circle amplification reaction was performed using 120 ng of genomic DNA isolated from $2 \times 10^5$ cells with QCP lysis buffer. DNA concentration was measured by Qubit BR Assay (Thermo Fisher Scientific #Q32853). For each experiment, reactions without the addition of Φ29 polymerase were included as a control as well as genomic DNA from ALT-positive (U2OS) and telomerase-positive (HeLa) cells. Cells were judged as ALT-positive if the signal intensity of the reaction with polymerase (+Φ29) subtract the intensity of the C-circle signal of the reaction without polymerase (-Φ29) from each respective sample was at least 5-fold higher than the background, in analogy to previous reports[12,13].

**Probe generation and hybridisation**. DIG-labelled probes were generated as described by either 3′-End-Labelling (Alu, C-circle telomere probe) or nick-translation (TRF telomere probe)[51] using commercial kits (ROCHE #03353575910 and #11745816910, respectively). For TRF Southern blot analysis, probes were generated by primer dimer PCR, which resulted in DNA of varying length made up of telomeric repeats. To this end, primers which contained telomeric repeats, were used for PCR in GoTaq master mix (Promega). The resulting amplicons were purified using PCR cleanup kit (Qiagen) and subjected to DIG nick-translation. 4 μg of denatured (100 °C for 10 min) 1 kb DNA ladder (NEB N3232) was used to generate marker-specific, DIG-labelled probe for TRF Southern blot using DNA Labeling Kit (ROCHE #11004760001). For the C-circle assay, probes were generated by DIG 3′-end-labelling oligonucleotides specific to Alu repeats or telomeric repeats with the kit described above.

All membranes were pre-hybridised in 5X SSC, 0.1% Sodium sarkosyl, 0.04% SDS for at least 1 h. Hybridisation was carried out overnight at 65 °C in the same buffer with 2.5 pmol/mL of 3′-end-labelling oligonucleotide probe or 12 ng/mL of nick translation-labelled telomeric probe. Membranes were washed three times with 2X SSC, 0.1% SDS and once with 2X SSC, all at room temperature and 15 min each. Subsequently, membranes were blocked in maleic acid buffer (100 mM maleic acid, 150 mM NaCl) containing 1% w/v blocking reagent (ROCHE #11096176001) for 30 min. Hybridised probes were detected with α-DIG (1:10,000 ROCHE #11093274910). Membranes were washed twice 15 min in maleic acid buffer supplemented with 0.3% Tween-20 and equilibrated in AP detection buffer (100 mM Tris-HCl pH 7.5, 100 mM NaCl), and signal was generated using CDP star substrate (ROCHE #12041677001).

**Western blot**. Western blot was performed as described[42]. Primary antibodies were α-RAD52 (Abcam ab124971, 1:2,000), α-BLM (Abcam ab2179, 1:1,000), α-PolD3 (Abcam ab182564, 1:5,000), α-SLX4 (kind gift by Prof. John Rousse, 1:5,000), α-ATRX (Bethyl A301-045A, 1:500), α-DAXX (Santa Cruz sc7152, 1;1,000), α-H3 (Abcam ab10799, 1:2,000), α-ASF1a (Cell Signalling 2990, 1:1,000), α-LANA (Millipore MABE1109, 1:1,000), α-ORF57 (Santa Cruz sc135746 1:1,000), α-beta actin (Biolegend 664802 1;2,000). Secondary HRP-coupled secondary antibodies were α-mouse (Dako P0447 1:2,000), α-rabbit (Dako P0399 1:2,000), α-rat (Abcam ab97057 1:2,000). Uncropped blots are available in Source data file.

**Reporting summary**. Further information on research design is available in the Nature Research Reporting Summary linked to this article.

## Data availability
Any additional source data will be made available upon reasonable request to the corresponding author. Source data are provided with this paper.

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

## Acknowledgements

We thank members of the Boulton laboratory for suggestions, discussions and critical reading of the manuscript. We thank members of the Ojala laboratory for providing key reagents during the course of the study. We thank Experimental Histopathology science technology platform (STP) of the Francis Crick Institute for providing healthy skin tissue for analysis and assistance with FFPE cell lines. We thank Prof. Justin Weir for assistance with sectioning the KS tumour biopsies. The work in the Boulton laboratory is supported by the Francis Crick Institute, which receives its core funding from Cancer Research UK (FC0010048), the UK Medical Research Council (FC0010048), and the Wellcome Trust (FC0010048); a European Research Council (ERC) Advanced Investigator Grant (Tel-Metab); and a Wellcome Trust Senior Investigator Grant. The work in the Feldhahn laboratory was supported by an Imperial College London PhD fellowship to T.L. and a Bennett Fellowship of Blood Cancer UK (formerly Bloodwise) to N.F. (P47415). P.J.F. was supported by MRC grant MR/S022597/1 and by NIHR Imperial BRC. P.M.O. was supported by the Academy of Finland Centre of Excellence grant (Translational Cancer Biology), and Sigrid Juselius Foundation, Finland.

## Author contributions

T.L., G.S., N.F. and S.J.B. conceived the study. T.L, G.S., N.F., P.M., A.I. and S.J.B designed experiments. P.O. provided KSHV-related reagents and technical advice. P.J.F. provided advice concerning study design and co-supervision. T.L., P.M. and G.S. conducted PICh analysis. T.L., P.M., A.V. and A.I evaluated hallmarks of ALT in cell lines and tumour biopsies. M.B. provided KS tumour biopsies and advice on the selection of clinical material. T.L. and S.J.B. wrote the manuscript with editorial input from P.M., A.I. and the other authors.

## Competing interests

The authors declare no competing interests.
