## [Peer Review File · Nature Communications]

REVIEWER COMMENTS

Reviewer #1 (Remarks to the Author):

In this manuscript Lippert et al show compelling evidence that KSHV infection promotes the alternative lengthening of telomeres (ALT) pathway of telomere maintenance. In multiple cell lines and clonal isolates latent KSHV infection promotes APBs, C-circles, an active telomere DNA damage response, telomere recombination and increased telomere length. Bystander and tumour samples from Kaposi Sarcoma patients show hallmarks of ALT in the cancer cells, but not adjacent non-cancer tissues. Further, they show that BIR components, which are known to drive ALT-dependent telomere elongation, promote proliferation of KSHV infected cells and contribute to KSHV genome maintenance.

I have one major consideration that is up to the discretion of the authors and editor on how to address. The remainder of my comments are minor suggestions to improve the manuscript. My recommendation is to invite a revised manuscript and publish upon adequate consideration of the reviewer's comments.

Major comment

ALT is defined as telomere maintenance in the absence of telomerase. Despite the titular claim, it is not demonstrated directly in this manuscript that KSHV infection triggers ALT per se. This needs to be addressed through experimentation or manuscript editing. Either are acceptable for publication in my opinion.

The data eloquently demonstrate KSHV infection drives ALT associated phenotypes. While these phenotypes strongly correlate with ALT activity, telomere maintenance in the absence of telomerase is not shown. Bona fide ALT activity is demonstrated through copying of a DNA tag inserted within a telomere to other chromosome ends (see examples from the Reddel and/or Pickett lab), or telomere length maintenance in the absence of telomerase. The authors can demonstrate KSHV drives ALT by showing evidence of telomere copying in infected cells; or by deleting telomerase (hTR or TERT) using CRISPR and demonstrating continued telomere maintenance in KSHV infected cells without telomerase activity. If shown, this would be a very exciting outcome.

Alternatively, the authors can change the title to "Oncogenic herpesvirus KSHV triggers the phenotypic hallmarks of alternative lengthening of telomeres" (or something similar). Even without direct identification of ALT activity, it is quite interesting that KSHV induces all the hallmarks of ALT. This will establish KSHV as a novel reagent to study telomere biology and opens a new and exciting avenue of enquiry on the unexpected influence of viral infection on chromosome ends. If the authors elect this route, they will need to expand the discussion to address what is demonstrated in this paper.

Should the authors elect to address this comment experimentally, and there is no evidence of ALT activity, this would also be interesting by showing a clear dissociation between ALT phenotypes and ALT activity. In this reviewer's opinion, it remains unclear which of the ALT phenotypes drive telomere extension and which are by-products of the molecular outcomes at ALT telomeres. This would also be a publishable result that could be expounded on in the discussion.

Minor comments

Discussion: With a transfer to Nature Communications, there is more space to expand the discussion. The authors should feel free to do this if so inclined. I find the requirement of BIR components for KSHV DNA genome and proliferation curious. It would be interesting to know the author's opinion if ALT is induced by KSHV as a mechanism to promote cellular immortality. Or if KSHV promotes BIR to enable viral replication, and ALT is a secondary outcome.

References: Similarly, with the added space in Nature Communications it would be reasonable to cite primary discoveries where possible. E.g.: Henson et al 2009 for c-circles; Cesare et al 2009 for

DNA damage response activation at ALT telomeres; Conomos et al 2012 for nuclear receptors at ALT telomeres; an ALT-related review or the original discovery instead of Hanahan and Weinberg; Yeager et al 1999 for APBs; Londoño-Vallejo et al 2004 for telomere recombination. I would also include Sobinoff et al 2017 for BIR in ALT. It might also be nice to add a BIR reference or review.

Line 141: TRAP assays are in-direct measure of telomerase activity.

Line 183 and figure 3g-i: the LANA/EdU Foci that overlap with telomeres, are these APBs? I expect so as viral DNAs, extrachromosomal telomere repeats, and telomeres accumulate in these structures. Also, is the BrdU signal telomere synthesis, or viral synthesis? Or both?

Line 384: Are biological replicates included in this experiment? Technical duplicates are not sufficient.

Line 399: I am confused why this is a paired t-test? My understanding is paired tests is when a measurement is taken before and after treatment. These would be un-paired measurements as the cells are not re-tested before and after infection.

Line 491: I was confused with what (1/3 PML bodies at telomeres mean)? What does the 1/3 refer to?

Figures: the red/green coloration maybe be difficult for some colorblind readers. Consider changing. This is particularly true for Figure 3i.

I applaud the inclusion of loading controls in the c-circle assays. This is typically ignored in these experiments. It is nice to see the experiment done well.

Reviewer #2 (Remarks to the Author):

In this manuscript Lippert et al., present on the interesting finding that the Oncogenic Herpesvirus KSHV induces the acquisition of ALT in previously non-ALT/telomerase positive cells and demonstrate that KSHV infection leads to sustained acquisition of the ALT pathway. The findings are certainly of great interest to the readership of Nature Communications and the data is of high quality overall, albeit with some controls missing. The authors clearly demonstrate that infection of KSHV triggers the ALT pathway although the mechanism by which this occurs remains largely unexplored which would greatly add to the publication. The finding that BIR is required for replication of the viral genome is, however, slightly less clear from the data presented and the authors may need to include a couple of additional experiments to strengthen this conclusion prior to publication. More detailed comments are given below:

In extended data 1 the authors provide a control to demonstrate that the KSHV infection is strictly latent at the point of PICH analysis. The PICH analysis was, however, performed in BJAB cells and the control shown in extended data 1 is for the SLK and EA.hy926 cell lines. This is at first reading confusing and the authors should include BJAB cells in this control.

Figures 3A and B require an accompanying immunoblot control of knockdown efficiency of each BIR factor to demonstrate an equivalent knockdown efficiency in both non transfected and viral transfected cells.

An accompanying immunoblot to assess knockdown efficiency is included to accompany figures 3C and E (Figure 3D). It is, however, not clear which cell line this was performed in and data should be included for both SLK and EA.hy926 cell lines as they were both included for analysis in Figure 3E.

For Figure 3E the authors demonstrate that knockdown of BIR factors leads to a reduction of viral episomes and conclude that the loss of BIR factors in the host cell reduces the ability of KSHV to establish latency upon de novo infection. While this is certainly plausible another explanation could be that, as loss of BIR factors reduce the survival of KSHV infected cells (Figure 3B), the cells that do survive and are subsequently analysed have a lower level of viral latency by selection, rather than that BIR itself is required for viral replication? This could be addressed using the assay in Figure 3F by doing the same experiment with depletion of a BIR factor to show that replication of the viral genome is directly abrogated upon loss of BIR activity.

From Figure 3F it is also difficult to definitively say from this data that the virus is specifically replicating in G2 without including other cell cycle phases.

In Figure 3h the authors need to be careful with their interpretation. Presumably loss of BIR factors should decrease EdU foci at telomeres due to loss of ALT related BIR. The authors show that the total number of LANA foci, however, does not decrease, suggesting that association of LANA with telomeres or generally into foci is independent of BIR. Taken together the evidence that the replication of the viral genome is dependent on telomeric BIR is purely correlative and would require some additional confirmation prior to publication.

To begin to explore potential mechanisms by which KSHV infection may trigger the ALT pathway the authors explore whether the levels of both ATRX or ASF1 are decreased upon infection of KSHV (Extended data Figure 8). The authors claim they see no change in level of ATRX, however, in the representative image it does appear that ATRX levels are decreased in the SLK cell line?

The manuscript could also be greatly strengthened by looking at the localisation of ATRX and ASF1 at telomeres upon KSHV transfection, either by IF or ChIP analysis, to give a more comprehensive insight as to whether the function of these proteins at telomeres is affected.

Minor Comments

The manuscript does not appear to include materials and methods (apologies if I've missed this?).

In figure 1, where the authors demonstrate an increase in ALT markers upon KSHV transfection via a variety of assays it would be valuable to include a known ALT positive control, such as U-2OS to gauge the extent of ALT activation.

For Figure 1e – it would be good to show a representative blot of the C-circles, at least in supplementary data.

In Line 114 the referencing of the relevant figure is wrong and should read Figure 1f,g. Likewise Line 115 should read Figure 1h, i.

The authors should be consistent with how they quantitate TIFs and APBs throughout the manuscript.

Reviewer #3 (Remarks to the Author):

Lippert et al. conclude that KSHV “triggers alternative lengthening of telomers” which is intriguing science. They provide a wide range of data to support their conclusion which, however, requires more detail for a reviewer or a reader to be able to evaluate. Without such detail, their conclusions are not adequately supported.

For example, they write “Telomerase positive BJAB cell lines with stable latent viral infection were established by de novo KSHV infection as described previously (15) and were analysed at 30 days post-infection in the absence of detectable lytic reactivation (Extended data Fig.1).” Reference 15 does not describe such de novo infection nor does (Extended data Fig.1) mention BJAB cells. In

fact, the authors apparently fail to provide data to show that the BJAB cells are infected with KSHV nor how many viral episomes are present per cell. Their FISH data are beautiful; why not measure the KSHV episomes by FISH?

In Figure 2, the authors measure Telomere mean intensity in AU, Telomere foci/nucleus (up to 200/ nucleus), Telomere length in AU without detailing how any of these measurements were accomplished. They need to do so.

The authors also provide data that appear to be inconsistent with their text. They write "Analysis of absolute viral copy number by qPCR revealed that depletion of BIR factors resulted in a decrease in the levels of viral episomes when compared to control siRNA (Fig.3e)." In no place do the authors provide "absolute copy number" which is needed to characterize their cells. For example, their extended Figure 6b would not allow them to measure numbers of less than 10 episomes per cell. They need to provide absolute copy numbers.

Finally, the authors use cell lines in culture that are already immortalized prior to exposure to KSHV. Their findings thus reflect an effect of KSHV in cells that have evolved to maintain their telomeres. The significance of their findings would be greatly enhanced if they would replicate them in primary cells such as HUVEC following infection with KSHV.

Reviewed by Bill Sugden

Reply to reviewer comments

Reviewer #1

In this manuscript Lippert et al show compelling evidence that KSHV infection promotes the alternative lengthening of telomeres (ALT) pathway of telomere maintenance. In multiple cell lines and clonal isolates latent KSHV infection promotes APBs, C-circles, an active telomere DNA damage response, telomere recombination and increased telomere length. Bystander and tumour samples from Kaposi Sarcoma patients show hallmarks of ALT in the cancer cells, but not adjacent non-cancer tissues. Further, they show that BIR components, which are known to drive ALT-dependent telomere elongation, promote proliferation of KSHV infected cells and contribute to KSHV genome maintenance.

I have one major consideration that is up to the discretion of the authors and editor on how to address. The remainder of my comments are minor suggestions to improve the manuscript. My recommendation is to invite a revised manuscript and publish upon adequate consideration of the reviewer's comments.

Major comment

ALT is defined as telomere maintenance in the absence of telomerase. Despite the titular claim, it is not demonstrated directly in this manuscript that KSHV infection triggers ALT per se. This needs to be addressed through experimentation or manuscript editing. Either are acceptable for publication in my opinion.

The data eloquently demonstrate KSHV infection drives ALT associated phenotypes. While these phenotypes strongly correlate with ALT activity, telomere maintenance in the absence of telomerase is not shown. Bona fide ALT activity is demonstrated through copying of a DNA tag inserted within a telomere to other chromosome ends (see examples from the Reddel and/or Pickett lab), or telomere length maintenance in the absence of telomerase. The authors can demonstrate KSHV drives ALT by showing evidence of telomere copying in infected cells; or by deleting telomerase (hTR or TERT) using CRISPR and demonstrating continued telomere maintenance in KSHV infected cells without telomerase activity. If shown, this would be a very exciting outcome.

Alternatively, the authors can change the title to "Oncogenic herpesvirus KSHV triggers the phenotypic hallmarks of alternative lengthening of telomeres" (or something similar). Even without direct identification of ALT activity, it is quite interesting that KSHV induces all the hallmarks of ALT. This will establish KSHV as a novel reagent to study telomere biology and opens a new and exciting avenue of enquiry on the unexpected influence of viral infection on chromosome ends. If the authors elect this route, they will need to expand the discussion to address what is demonstrated in this paper.

Should the authors elect to address this comment experimentally, and there is no evidence of ALT activity, this would also be interesting by showing a clear dissociation between ALT phenotypes and ALT activity. In this reviewer's opinion, it remains unclear which of the ALT phenotypes drive telomere extension and which are by-products of the molecular outcomes at ALT telomeres. This would also be a publishable result that could be expounded on in the discussion.

We agree that determining the extent to which telomere elongation remains dependent on telomerase activity in KSHV-infected cells is an interesting experiment. However, all recorded incidences of ALT induction using initially telomerase positive cell lines^{1,2} are concomitant with residual telomerase activity in the TRAP assay. This may reflect ALT activation in the established genetic background of a telomerase-immortalised cell line. In our study, the mean increased and heterogeneous changes in telomere length (Fig.2h) recorded here occur together with reduced *hTERT* expression and active

telomerase protein in cell lysate of infected cells (Fig.2i,j). Therefore, while we cannot unequivocally rule out a contribution of telomerase to the phenotype recorded here, we are confident that the telomere length increase observed here is not due to telomerase hyperactivation.

The first experiment suggested by the reviewer here, eg. copying of a tag to homologous chromosomes, as initially established by Roger Reddel, is analogous to the CO-FISH analysis presented in Fig. 2a,b. With this assay, we prove the same point: KSHV infection increases the incidence of crossing-over events by recombination at telomeres together with reduced *hTERT* expression.

The reviewer rightly points out that the only way to address the major comment in a thorough manner would be the generation of KSHV-infected, telomerase knockout cell lines. Given that this would encompass generation and validation of multiple cell lines, loss-of-function validation, and phenotypic analysis, we concur with the reviewer that while conclusive, these experiments are outside the scope of the current study.

Therefore, in light of the comments made by Reviewer #1, we agree to change the title of the current study to “Oncogenic herpesvirus KSHV triggers the hallmarks of alternative lengthening of telomeres” to more accurately reflect the implications of our data. We have also expanded and modified the discussion section in the manuscript to reflect this change.

Minor comments

Discussion: With a transfer to Nature Communications, there is more space to expand the discussion. The authors should feel free to do this if so inclined. I find the requirement of BIR components for KSHV DNA genome and proliferation curious. It would be interesting to know the author’s opinion if ALT is induced by KSHV as a mechanism to promote cellular immortality. Or if KSHV promotes BIR to enable viral replication, and ALT is a secondary outcome.

We completely agree with the reviewer’s comment here and have now included a more comprehensive discussion section, which covers the points made in the above paragraph.

Experimentally, we have now performed BrdU pull-down assays in G2 synchronised cells to assess whether KSHV replication takes place predominantly via BIR. However, contrary to this notion, our data clearly show that KSHV replication is not affected by loss of BIR factors in cells (Fig. 3f). However, knockdown of BIR factors significantly decreases the ability of KSHV to establish latency upon infection *de novo* (Fig. 3c-e). Therefore, on the basis of these new data, we suggest a model where KSHV requires intact BIR to establish latency and ALT activation is a by-product of KSHV localization to telomeric sites.

References: Similarly, with the added space in Nature Communications it would be reasonable to cite primary discoveries where possible. E.g.: Henson et al 2009 for c-circles; Cesare et al 2009 for DNA damage response activation at ALT telomeres; Conomos et al 2012 for nuclear receptors at ALT telomeres; an ALT-related review or the original discovery instead of Hanahan and Weinberg; Yeager et al 1999 for APBs; Londoño-Vallejo et al 2004 for telomere recombination. I would also include Sobinoff et al 2017 for BIR in ALT. It might also be nice to add a BIR reference or review.

We followed the reviewer’s advice and added all of the additional references. We also added one additional reference to a review on the mechanism of BIR.

Line 141: TRAP assays are in-direct measure of telomerase activity.

We agree and have removed the word “direct” in this sentence to account for this.

Line 183 and figure 3g-i: the LANA/EdU Foci that overlap with telomeres, are these APBs? I expect so as viral DNAs, extrachromosomal telomere repeats, and telomeres accumulate in these structures. Also, is the BrdU signal telomere synthesis, or viral synthesis? Or both?

We agree with the reviewer’s assessment. BIR-associated telomere/EdU foci in G2/M are commonly associated with PML bodies³. The question whether these EdU signals represent active telomere DNA synthesis or viral DNA synthesis is in principle still outstanding. Given that viral replication seems largely independent of canonical factors involved in BIR DNA synthesis (Fig.3 f), we favour the former scenario. However, we still think that either possibility may theoretically occur here, and both are discussed in greater detail in the modified discussion section of the manuscript.

Line 384: Are biological replicates included in this experiment? Technical duplicates are not sufficient.

These experiments were performed in biological duplicate on the basis of cell lines established for previous reports by Prof. Thomas Schulz, Hanover Medical School in collaboration with Paivi Ojala, University of Helsinki⁴. We apologise for this error and have corrected this in the modified version of the manuscript.

Line 399: I am confused why this is a paired t-test? My understanding is paired tests is when a measurement is taken before and after treatment. These would be un-paired measurements as the cells are not re-tested before and after infection.

We think that in this case, the conditions to apply a paired t-test are met since the cells are infected with KSHV and compared to uninfected, co-cultures cells. Nevertheless, we performed unpaired statistical analysis using the Mann-Whitney test as well. In all cases, given the fixed sample size, the outcome of the test $p=0.0286$ reaches significance as it satisfies the $p<0.1$ threshold.

Line 491: I was confused with what (1/3 PML bodies at telomeres mean)? What does the 1/3 refer to?

1/3 refers to one third of total PML bodies at telomeres. This is to account for cell-type-intrinsic differences in the number of PML bodies within each patient tumour’s tissue section. We apologize for not making this clear in the submitted manuscript. This is now made explained in the modified version of the manuscript as well as in the materials and methods section, where more detail on automated image analysis employed may be found.

Figures: the red/green coloration maybe be difficult for some colorblind readers. Consider changing. This is particularly true for Figure 3i.

Thank you for this suggestion. We have changed the colour scheme of all figures to make them accessible for people suffering from colour blindness.

I applaud the inclusion of loading controls in the c-circle assays. This is typically ignored in these experiments. It is nice to see the experiment done well.

We concur with the notion that this experiment is often performed without this vital control in the literature.

Reviewer #2 (Remarks to the Author):

In this manuscript Lippert et al., present on the interesting finding that the Oncogenic Herpesvirus KSHV induces the acquisition of ALT in previously non-ALT/telomerase positive cells and demonstrate that KSHV infection leads to sustained acquisition of the ALT pathway. The findings are certainly of great interest to the readership of Nature Communications and the data is of high quality overall, albeit with some controls missing. The authors clearly demonstrate that infection of KSHV triggers the ALT pathway although the mechanism by which this occurs remains largely unexplored which would greatly add to the publication. The finding that BIR is required for replication of the viral genome is, however, slightly less clear from the data presented and the authors may need to include a couple of additional experiments to strengthen this conclusion prior to publication. More detailed comments are given below:

In extended data 1 the authors provide a control to demonstrate that the KSHV infection is strictly latent at the point of PICh analysis. The PICh analysis was, however, performed in BJAB cells and the control shown in extended data 1 is for the SLK and EA.hy926 cell lines. This is at first reading confusing and the authors should include BJAB cells in this control.

We have now performed this control experiment for BJAB cells as well, see Extended data Figure 1b.

Figures 3A and B require an accompanying immunoblot control of knockdown efficiency of each BIR factor to demonstrate an equivalent knockdown efficiency in both non transfected and viral transfected cells.

An accompanying immunoblot to assess knockdown efficiency is included to accompany figures 3C and E (Figure 3D). It is, however, not clear which cell line this was performed in and data should be included for both SLK and EA.hy926 cell lines as they were both included for analysis in Figure 3E.

These control blots have now been added in Extended data Figure 5a.

For Figure 3E the authors demonstrate that knockdown of BIR factors leads to a reduction of viral episomes and conclude that the loss of BIR factors in the host cell reduces the ability of KSHV to establish latency upon de novo infection. While this is certainly plausible another explanation could be that, as loss of BIR factors reduce the survival of KSHV infected cells (Figure 3B), the cells that do survive and are subsequently analysed have a lower level of viral latency by selection, rather than that BIR itself is required for viral replication? This could be addressed using the assay in Figure 3F by doing the same experiment with depletion of a BIR factor to show that replication of the viral genome is directly abrogated upon loss of BIR activity.

We have now included this experiment (Fig 3 f, Extended data Fig.7a,b) and show that viral genome synthesis in G2/M is largely independent of intact BIR. On the basis of this experiment, we suggest that BIR may be important for viral latency in a manner that does not involve bulk synthesis of KSHV DNA. Please see the modified discussion for details.

From Figure 3F it is also difficult to definitively say from this data that the virus is specifically replicating in G2 without including other cell cycle phases.

This is an intrinsic challenge of this experiment. Residual S-phase cells are included in such analysis by default and this has been documented in applications of this technique by other laboratories. In order to evaluate the contribution of synthesis of viral DNA during S-phase in the context of our assay, we performed an appropriate control presented in Extended data Fig.7c,d. Indeed, BrdU incorporation into

KSHV episomes is not limited to G2 as it is highly prevalent in S phase, which again argues that BIR is not the main mode of viral DNA synthesis as this is believed to occur in G2/M phase.

In Figure 3h the authors need to be careful with their interpretation. Presumably loss of BIR factors should decrease EdU foci at telomeres due to loss of ALT related BIR. The authors show that the total number of LANA foci, however, does not decrease, suggesting that association of LANA with telomeres or generally into foci is independent of BIR. Taken together the evidence that the replication of the viral genome is dependent on telomeric BIR is purely correlative and would require some additional confirmation prior to publication.

We have now performed an assay measuring BrdU incorporation into KSHV episomes during G2 phase and examined the effect of knockdown of BIR factors in this context. Our results show that KSHV replication is not dependent on intact BIR (Fig.3f). We have changed the text in manuscript accordingly.

To begin to explore potential mechanisms by which KSHV infection may trigger the ALT pathway the authors explore whether the levels of both ATRX or ASF1 are decreased upon infection of KSHV (Extended data Figure 8). The authors claim they see no change in level of ATRX, however, in the representative image it does appear that ATRX levels are decreased in the SLK cell line?

We were also interested in analysing this change in more detail. This has now been included in Extended Figure 9. Indeed, ATRX levels are moderately decreased in EA.hy926 cells at both the protein and mRNA level. In SLK cells, there was a small but statistically insignificant reduction in ATRX mRNA and marginal decrease in ATRX protein upon infection. DAXX remained unchanged in EA.hy926 cells upon infection, whereas we detected an increase in both the protein and mRNA level upon infection in SLK cells. We also examined ASF1a protein levels and found that they remained unchanged upon infection with KSHV. However, we detected a statistically significant decrease in ASF1a transcripts upon infection. Taken together, we do indeed detect modest changes in the levels of ATRX, DAXX, and ASF1a. However, we do not think that these differences are sufficient to consider them functional triggers of the hallmarks of ALT documented in this report.

The manuscript could also be greatly strengthened by looking at the localisation of ATRX and ASF1 at telomeres upon KSHV transfection, either by IF or ChIP analysis, to give a more comprehensive insight as to whether the function of these proteins at telomeres is affected.

Given that we provide analysis at the protein and mRNA level in the revised manuscript, we think that further analysis of this phenomenon is not within the scope of this study. We agree with the Reviewer that the suggested experiment would be informative, however localisation of ATRX or ASF1 to telomeres has not been shown in the literature on differentiated human cells by IF or ChIP, only using proteomics analysis. In order to evaluate the contribution of these proteins in the induction of ALT hallmarks upon KSHV infection, reagents which permit the analysis suggested here would have to be generated. In addition to this, the function of these proteins may be modulated in ways other than their localisation to telomeres and this would have to be addressed with detailed investigation.

Minor Comments

The manuscript does not appear to include materials and methods (apologies if I've missed this?).

We apologise for this error. This occurred due to the transfer from nature cancer, where materials and methods are submitted as a separate document, which is evidently not automatically included upon

transfer to nature communications. We were unaware of this. The final submission now includes a detailed materials and methods section.

In figure 1, where the authors demonstrate an increase in ALT markers upon KSHV transfection via a variety of assays it would be valuable to include a known ALT positive control, such as U-2OS to gauge the extent of ALT activation.

ALT cell lines vary in the extent to which they exhibit each ALT hallmarks. To give an indication of how the KSHV-induced ALT compares to established cell lines using ALT for telomere maintenance, we compared key hallmarks of ALT to U2OS, one of the “strongest” ALT cell lines, as requested and include this in Extended figure 4. As expected, U2OS consistently exhibits more drastic but comparable telomeric phenotypes than KSHV-infected EA.hy926 and SLK in key ALT indicators such as TRF Southern blotting, C-Circle analysis, T-SCE, and APB analysis (Extended figure 4a-e). Given that our KSHV-infected cell lines exhibit residual telomerase activity and were established from initially non-ALT precursor cell lines, we think that this comparison to ALT tumour-derived U2OS is in line with expectations under these circumstances.

For Figure 1e – it would be good to show a representative blot of the C-circles, at least in supplementary data.

We agree that this is needed and have added a representative image in Figure 1e.

In Line 114 the referencing of the relevant figure is wrong and should read Figure 1f,g. Likewise Line 115 should read Figure 1h, i.

Thank you for spotting this error, we have corrected this in the revised manuscript.

The authors should be consistent with how they quantitate TIFs and APBs throughout the manuscript.

The quantification of TIFs and APBs is now consistent for the analysis of cell lines (Fig.1c,g,i) and tumour tissue (Fig. 4f,g) respectively. All experiments performed with cell lines are quantified using the same thresholds for each stain. However, signals obtained from tumour biopsies were significantly different from cell lines. We also recorded variation between signal from different patients (Extended figure 11a) and cell-type intrinsic differences, which required normalisation for unbiased analysis (Extended figure 10b-d). We therefore deem it appropriate to apply different thresholds for automated co-localisation analysis *in vitro* and *in vivo*.

Reviewer #3 (Remarks to the Author):

Lippert et al. conclude that KSHV “triggers alternative lengthening of telomers” which is intriguing science. They provide a wide range of data to support their conclusion which, however, requires more detail for a reviewer or a reader to be able to evaluate. Without such detail, their conclusions are not adequately supported.

For example, they write “Telomerase positive BJAB cell lines with stable latent viral infection were established by de novo KSHV infection as described previously (15) and were analysed at 30 days post-infection in the absence of detectable lytic reactivation (Extended data Fig.1).” Reference 15 does not describe such de novo infection nor does (Extended data Fig.1) mention BJAB cells. In fact, the authors apparently fail to provide data to show that the BJAB cells are infected with KSHV nor how many viral episomes are present per cell.

Similar issues were also raised by Reviewer #2. We now include appropriate Western blot analysis addressing these concerns. This is now included in Extended Figure 1b of the revised manuscript.

Their FISH data are beautiful; why not measure the KSHV episomes by FISH?

We had technical difficulties when attempting to detect KSHV episomes by FISH. As an alternative, we now provide extensive analysis of KSHV episome replication using pulldown and ddPCR approaches in Fig. 3f, Extended data Fig. 7a,b, which allow analysis of KSHV episomes in infected cells in a quantitative manner analogous to FISH. We were also able to evaluate the dependency of intact KSHV replication on factors involved in ALT.

In Figure 2, the authors measure Telomere mean intensity in AU, Telomere foci/nucleus (up to 200/nucleus), Telomere length in AU without detailing how any of these measurements were accomplished. They need to do so.

Apologies for omitting this information. We have now clarified this in the materials and methods section and also included this in the figure legend.

The authors also provide data that appear to be inconsistent with their text. They write “Analysis of absolute viral copy number by qPCR revealed that depletion of BIR factors resulted in a decrease in the levels of viral episomes when compared to control siRNA (Fig.3e).” In no place do the authors provide “absolute copy number” which is needed to characterize their cells. For example, their extended Figure 6b would not allow them to measure numbers of less than 10 episomes per cell. They need to provide absolute copy numbers.

We modified the description in the revised text as “Analysis of relative viral copy number” throughout, which addresses this concern.

Finally, the authors use cell lines in culture that are already immortalized prior to exposure to KSHV. Their findings thus reflect an effect of KSHV in cells that have evolved to maintain their telomeres. The significance of their findings would be greatly enhanced if they would replicate them in primary cells such as HUVEC following infection with KSHV.

Reviewed by Bill Sugden

We agree that the uninfected progenitor SLK and EA.hy926 cells, which we use as a basis to generate KSHV-infected cell lines, have evolved to maintain their telomeres (by telomerase, see also comments made in response to Reviewer #1). In this regard, we agree that our experiments do not allow examination of the significance of ALT induction for immortalization of cells by KSHV. However, in contrast to EBV

infection, KSHV does not readily immortalize primary cells in culture and such experimental systems often include hTERT activation to aid immortalization, for instance, by co-expression of papillomavirus E6⁵. In this study, we instead utilize a cohort of primary tumour material from KS patients to examine the end point of KSHV-induced cell immortalization and find evidence for ALT activity.

Therefore, we appreciate that the significance of these findings for the mechanism of cell immortalization by KSHV in the context of KS are limited. Indeed, several important questions about the genesis of KS tumour cells remain in the field such as precise cell of origin, whether KS is truly an immortalized neoplasm, and the role of KSHV re-infection and lytic replication. We cannot address these in the current study. What we do show clearly on the basis of our analysis of initially hTERT-immortalized cell lines and primary tumour tissue, is that KSHV triggers all hallmarks of ALT. While this does not permit the study of cell immortalization by KSHV, it does provide a tool to study induction of ALT, which is the most significance aspect of this report.

Additional control experiments added to revised manuscript:

We now add a cohort of healthy skin donors to confirm the induction of C-circles, which we observe in KS tumour tissue. Indeed, we detect a statistically significant increase in the tumour cohort relative to healthy control (Fig. 4j). None of the healthy skin controls exceed the background of the assay and are therefore considered negative for C-circles (Extended data Fig. 11b).

References

1. Hu Y, Shi G, Zhang L, et al. Switch telomerase to ALT mechanism by inducing telomeric DNA damages and dysfunction of ATRX and DAXX. *Sci Rep.* 2016;6:32280. doi:10.1038/srep32280
2. O'Sullivan RJ, Arnoult N, Lackner DH, et al. Rapid induction of alternative lengthening of telomeres by depletion of the histone chaperone ASF1. *Nat Struct Mol Biol.* 2014;21(2):167-174. doi:10.1038/nsmb.2754
3. Zhang J-M, Yadav T, Ouyang J, Lan L, Zou L. Alternative Lengthening of Telomeres through Two Distinct Break-Induced Replication Pathways. *Cell Rep.* 2019;26(4):955-968.e3. doi:10.1016/j.celrep.2018.12.102
4. Cheng F, Weidner-Glunde M, Varjosalo M, et al. KSHV Reactivation from Latency Requires Pim-1 and Pim-3 Kinases to Inactivate the Latency-Associated Nuclear Antigen LANA. *PLOS Pathog.* 2009;5(3):e1000324. <https://doi.org/10.1371/journal.ppat.1000324>.
5. McAllister SC, Hanson RL, Grissom KN, Botto S, Moses A V. An In Vitro Model for Studying Cellular Transformation by Kaposi Sarcoma Herpesvirus. *J Vis Exp.* 2017;(126). doi:10.3791/54828

REVIEWERS' COMMENTS

Reviewer #1 (Remarks to the Author):

The authors have addressed the reviewers comments appropriately and the manuscript is ready for publication. Congratulations to the authors.

Reviewer #2 (Remarks to the Author):

Lippert et al., have significantly strengthened the manuscript and addressed my primary concern regarding the interpretation that BIR factors may be required for replication of the viral genome. Indeed, the authors have now shown that this is not the case and suggest that BIR may be important for viral latency in a manner that does not involve bulk synthesis of viral DNA. I agree with this interpretation.

The authors have also largely addressed my other concerns. One very minor point is that in the new immunoblot included in Extended Figure 5a the knockdown of SLX4 is difficult to interpret due to the quality of the blot and the authors may wish to update this if they have a clearer blot.

Overall, I am delighted to now fully support the publication of this highly interesting article in Nature Communications.

Best Wishes,

David Clynes

Reviewer #3 (Remarks to the Author):

The authors have generally, nicely addressed the concerns of the reviewers. They need, though, to modify their legend to figure 3e to be in accord with the the figure itself. The legend states "Quantification of absolute KSHV episome copy number in cells..." which is incorrect while the figure axis describes "Relative number of KSHV genomes/cell" which the figure depicts.

Bill Sugden

Reviewer #1 (Remarks to the Author):

The authors have addressed the reviewers comments appropriately and the manuscript is ready for publication. Congratulations to the authors.

Thank you very much.

Reviewer #2 (Remarks to the Author):

Lippert et al., have significantly strengthened the manuscript and addressed my primary concern regarding the interpretation that BIR factors may be required for replication of the viral genome. Indeed, the authors have now shown that this is not the case and suggest that BIR may be important for viral latency in a manner that does not involve bulk synthesis of viral DNA. I agree with this interpretation.

The authors have also largely addressed my other concerns. One very minor point is that in the new immunoblot included in Extended Figure 5a the knockdown of SLX4 is difficult to interpret due to the quality of the blot and the authors may wish to update this if they have a clearer blot.

Overall, I am delighted to now fully support the publication of this highly interesting article in Nature Communications.

Best Wishes,

David Clynes

We also agree that while viral DNA synthesis by BIR was a tempting hypothesis based on our data, subsequent experiments clearly showed that this was not the case. We do not have a better blot for SLX4 unfortunately and will proceed with the existing data. Thank you for your interest in the study and your support for publication.

Reviewer #3 (Remarks to the Author):

The authors have generally, nicely addressed the concerns of the reviewers. They need, though, to modify their legend to figure 3e to be in accord with the the figure itself. The legend states "Quantification of absolute KSHV episome copy number in cells..." which is incorrect while the figure axis describes "Relative number of KSHV genomes/cell" which the figure depicts.

Bill Sugden

Thank you very much. We have changed the text in accordance with your comment.